# Content of Lead and Cadmium in the Tissues and Organs of the Wild Mallard Duck (*Anas platyrhynchos* L.) Depending on the Region of Poland Where It Is Harvested

**DOI:** 10.3390/ani13213327

**Published:** 2023-10-26

**Authors:** Elżbieta Bombik, Antoni Bombik, Katarzyna Pietrzkiewicz

**Affiliations:** Faculty of Agricultural Sciences, University in Siedlce, Prusa Street 14, 08-110 Siedlce, Poland; antoni.bombik@uph.edu.pl (A.B.); pietrzkiewicz.ka@gmail.com (K.P.)

**Keywords:** mallard duck, breast muscle, leg muscle, liver, environment pollution, lead, cadmium

## Abstract

**Simple Summary:**

A qualitative analysis of the tissues and organs of mallard ducks (*Anas platyrhynchos* L.) allows us to estimate the possible hygienic and toxicological threats related to the concentration of lead and cadmium in the environment. A common property of heavy metals is that even in low concentrations they can be toxic to living organisms, including humans. Research related to monitoring the presence of pollutants in the environment of free-living animals should be conducted in order to assess the degree of environmental contamination and issue opinions in this regard.

**Abstract:**

A property common to heavy metals is that even in small concentrations they can exert toxic effects on living organisms, including humans. The aim of this study was to analyze the quality of the tissues and organs of wild mallard ducks (*Anas platyrhynchos* L.) by estimating the potential hygiene and toxicological hazard associated with the concentrations of lead and cadmium in these tissues and organs, as well as the influence of the birds’ sex and place of origin on these parameters. A significantly higher average lead content was noted in the breast muscles and the livers of the mallards harvested in the Leszno hunting district compared to the birds from the Siedlce hunting district. A higher average cadmium concentration was recorded in the breast and leg muscles of the mallards harvested in the Leszno hunting district than the Siedlce hunting district. The concentration of cadmium in the tissues and organs of the mallards harvested in the Siedlce or the Leszno hunting district did not exceed the acceptable limits for the muscles and livers of slaughtered animals. This study found no significant effect of the sex of the wild crossbreeds on the content of lead and cadmium in tissues and organs. This study showed that the mallard has a measurable response to environmental pollution with lead and cadmium, and therefore it is a species that could to some extent be used as a bioindicator of the level of contamination of the environment with these xenobiotics. The ban on the use of lead pellets as ammunition in Poland may largely limit the degree of lead contamination of the tissues and organs of mallards.

## 1. Introduction

A property common to heavy metals is that even in small concentrations they can exert toxic effects on living organisms, including humans. These substances have carcinogenic and mutagenic effects and can lead to the development of diseases of civilization. The elements cadmium and lead are particularly toxic for living organisms [1].

Lead (Pb, *Lat. plumbum*) is a widespread element in the Earth’s crust and one of the most hazardous xenobiotics present in the environment. In natural conditions, it enters the environment mainly as a result of volcanic eruptions and rock weathering. However, its predominant source in the environment is anthropogenic pollution, e.g., from the mining industry and processing of lead ores and other nonferrous metals containing an admixture of lead, as well as from transport. Lead can also enter the soil together with mineral fertilizers and fertilizer lime. In Poland, according to the Regulation of the Ministry of Agriculture and Rural Development of 18 June 2008 [2] on the implementation of certain provisions of the Act on fertilizers and fertilization, the acceptable limit value of lead is 140 mg/kg d.w. of organic, organic and mineral and mineral fertilizer. The level of Pb impurities in fertilizer lime may not exceed 200 mg/kg of calcium oxide, while in the case of fertilizer lime containing magnesium, the permissible limit is 600 mg/kg of the sum of calcium oxide and magnesium oxide. Pb accumulates in the food chain, mainly as a result of human economic activity. Following ingestion, it reaches the bloodstream, where it binds to plasma proteins. Then, together with the peripheral blood, it reaches the heart, lungs, liver and kidneys, where it undergoes bioaccumulation. It subsequently accumulates in the skin and bones. In vertebrates, depending on the site of accumulation, it is excreted at varying rates. It is eliminated most quickly from the blood and parenchymal organs (a half-life of about 30 days) and most slowly from bone tissue (a half-life of up to 30 years) [3].

The presence of cadmium and mercury enhances the toxic effects of Pb [4]. Waterfowl poisoned with Pb can be a reservoir of viruses posing a threat to humans, such as avian flu [5]. In addition, the consumption of the meat of these birds poses a serious threat to human health. For many years, one of the main causes of Pb poisoning in waterfowl has been the swallowing of hunting pellets by the birds [6,7,8,9]. Intensive hunting causes the contamination of the aquatic environment with Pb from pellets. Pb pellets are swallowed accidentally by foraging birds or are mistaken for small stones and swallowed as gastroliths, which facilitate digestion. It is estimated that 15,000 to 50,000 tons of Pb are shot into the environment in Europe every year by hunters [7,8]. Kim and Oh [10] report that the amount of Pb fired into the environment in Korea exceeds 200 tons. The annual global losses in avifauna from this cause may affect several million individuals, which is one of the main reasons for total bans on the use of Pb ammunition or its exclusion in wetland areas in countries such as Denmark, the Netherlands, Norway or the United Kingdom [9,11].

Digestive juices cause Pb to form easily soluble compounds, which are absorbed in the intestines [8,12]. Pb poisoning in birds has a detrimental effect on their physiological condition, leading to developmental disorders of the central nervous system, reduced immunity, kidney failure, weakness, behavioral changes, diarrhea, anorexia and death [13,14]. Furthermore, reduced immunity due to Pb poisoning increases susceptibility to parasites [15,16].

According to the European Chemicals Agency, the use of Pb ammunition in wetland areas creates a risk for waterfowl because the birds swallow the used Pb pellet and Pb causes toxicological effects, including death. The number of waterbirds that die of Pb poisoning in the European Union is estimated at about a million per year. The use of Pb ammunition also poses a risk to animal species that feed on birds contaminated by it, as well as to people consuming birds shot with Pb ammunition. The agency has proposed that restrictions on the use of Pb ammunition should apply not only in wetlands but also in a 100-meter buffer zone around them [17]. Currently, the law in Poland is being adjusted to the recommendations of the European Union in this respect.

Cadmium (Cd, *Lat. cadmium*) is easily volatilized and dispersed in the environment, usually in the form of Cd oxide, which is easily soluble in water [18]. This favors the mobility and bioavailability of Cd in the environment.

In natural conditions, Cd enters the environment mainly due to volcanic eruptions. However, natural emissions of this element to the atmosphere account for only about 10% of its total emissions [4]. Cd reaches the environment primarily as a result of anthropogenic pollution, e.g., from mining, metallurgy and the energy industry (the combustion of hard coal). It is also used in industry for the production of nickel-Cd alkaline batteries, pigments, fluorescent paint, solders and alloys, Cd rods, plastic stabilizers and fireworks. Another important source of Cd is agriculture and the use of phosphate fertilizers. The phosphorites used to produce them are contaminated with Cd in the amount of 10–100 mg·kg^−1^ [19]. The long-term use of phosphate fertilizers causes persistent contamination of the soil with this metal. Cichy et al. [20] report that the average level of Cd in fertilizers sold in the EU is 45 mg Cd/kg P_2_O_5_. In Poland, according to the Regulation of the Ministry of Agriculture and Rural Development of 18 June 2008 [2] on the implementation of certain provisions of the Act on fertilizers and fertilization, the limit value of Cd is 5 mg/kg d.w. of organic or organic and mineral fertilizer and 50 mg/kg d.w. of mineral fertilizer. The level of Cd impurities in fertilizer lime may not exceed 8 mg/kg of calcium oxide, while in fertilizer lime containing magnesium, the permissible limit is 15 mg/kg for the sum of calcium oxide and magnesium oxide. Cd is taken up by aquatic fauna and flora in proportion to its concentration in the water and is easily incorporated in the food chain [4]. Once in the ecosystem, Cd is not degraded and remains in constant circulation [21]. Its long half-life of 10–30 years directly translates to accumulation in plants, animals and humans [22]. Cd is extremely harmful for humans and animals. Even small quantities are highly toxic. It mainly damages the liver, kidneys, bones and testicles, in which it is easily accumulated. However, the target organ accumulating this element is the kidneys [23,24]. Studies have shown that the exposure of birds to Cd causes a deterioration in their physiological condition, a reduction in their reproductive capacity and breeding outcomes, a deterioration in egg quality characteristics and disturbances in embryogenesis [25,26,27,28].

The law in Poland does not regulate the concentrations of heavy metals in the edible tissues and organs of game animals. For this reason, to assess the degree of contamination of the tissues and organs of mallard ducks, it seems reasonable to use the permissible limits for the concentration of heavy metals in the tissues and organs of animals raised for slaughter. The main legal provisions at the EU level defining the maximum levels of heavy metals in foodstuffs are Commission Regulation (EU) No. 488/2014 [29] and Commission Regulation (EU) No. 2015/1005 [30]. According to these regulations, the maximum permissible level of Pb contamination in meat (beef, poultry, mutton and pork) is 0.10 mg·kg^−1^ wet weight, while the level for offal from these species is 0.50 mg·kg^−1^ w.w. The upper limit for Cd in beef, pork, poultry and mutton is 0.05 mg·kg^−1^ w.w., and the limit for the liver of these animals is 0.50 mg·kg^−1^ w.w.

Game birds in Poland include four species of wild ducks: *Anas platyrhynchos* L., *Anas crecca* L., *Aythya ferina* L. and *Aythya fuligula* L. During the 2021/2022 hunting season, 45,235 wild ducks were culled, including 5035 in the Masovian Voivodeship and 4217 in the Greater Poland Voivodeship [31]. The farm breeding of mallards is carried out by Game Breeding Centers.

Raw meat from game animals, in addition to benefits such as high digestibility, nutritional value and protein content; low fat content; and a favorable ratio of unsaturated to saturated fatty acids, is also a valuable source of vitamins and minerals: sodium, potassium, calcium and zinc. The protein content in the muscles of wild mallards ranges from 19% to 21.8%, the fat content ranges from 3% to 6.1% and the caloric value ranges from 140 to 155 kcal [32]. According to Bombik et al. [33], given recommendations to limit the intake of saturated fatty acids in the human diet, the low proportion of these acids in the breast muscle of mallards may suggest that the meat of wild mallards is of higher nutritional value than that of farm animals. The higher proportion of monounsaturated fatty acids than saturated fatty acids in the breast and leg muscles of mallards is also beneficial for consumer health. Free-living animals search for food in their environment themselves. Game animals living in areas with higher emissions of heavy metal than in other regions accumulate them in their tissues in much greater amounts, which can pose a toxicological hazard [23].

The aim of this study was to analyze the quality of the tissues and organs of wild mallard ducks (*Anas platyrhynchos* L.) by estimating the potential hygiene and toxicological hazards associated with the concentrations of Pb and Cd in these tissues and organs, as well as the influence of the birds’ sex and place of origin on these parameters. Given the inadequate level of knowledge of the load of heavy metals in the Leszno and Siedlce hunting districts and the scarcity of qualitative analyses of the tissues and organs of wild mallards, an assessment was made on the suitability of the mallard as a bioindicator of the degree of environmental contamination with Cd and Pb.

## 2. Materials and Methods

### 2.1. Animals and Sample Collection

This study was carried out on material obtained from wild mallard ducks harvested in 2018–2019 in two study areas. The first area was the Siedlce hunting district, located in the central–eastern part of Poland, and the other was the Leszno hunting district, situated in the western part of Poland [33]. The two areas differed in environmental resources and in the degree of contamination of the environment with heavy metals. Due to differences in the level of crop production between the Masovian and Greater Poland Voivodeships, there are substantial differences in the consumption of mineral and lime fertilizers in the two regions.

The Siedlce hunting district is located in the Masovian Voivodeship. Its total area is 736,184 ha, with 556,423 ha occupied by fields and 179,761 ha occupied by forests [34]. The agrifood industry plays a dominant role in the production structure of the Masovian Voivodeship, with well-developed chemical, electrical machinery and energy industries as well. In the 2015/2016 business year, the consumption of mineral fertilizers (NPK) in the Masovian Voivodeship amounted to 217,500 t, including 118,320 t of nitrogen fertilizer (N), 38,420 t of phosphorus fertilizer (P_2_O_5_) and 60,680 t of potassium fertilizer (K_2_O), while the consumption of calcium fertilizer (CaO) amounted to 78,300 t [35]. In 2015, the amount of waste produced and stored in landfills in the Masovian Voivodeship was 7,096,200 t, and emissions of heavy metals to the atmosphere from the most harmful plants were 0.027 t for Cd and 1.221 t for Pb [36].

The Leszno hunting district is situated in southwestern Greater Poland. Its total area is 343,291 ha, including 273,396 ha of fields and 69,895 ha of forests [37]. Due to intensive crop production, the level of mineral fertilization in the Greater Poland Voivodeship in the 2015/2016 business year was 23.4% higher than the national average, and the level of calcium fertilizers was nearly 300% higher. The consumption of mineral fertilizers (NPK) amounted to 278,500 t, including 149,500 t of nitrogen fertilizer (N), 44,900 t of phosphorus fertilizer (P_2_O_5_) and 84,000 t of potassium fertilizer (K_2_O), while the consumption of calcium fertilizer (CaO) was 357,500 t [38]. The agrifood industry is dominant in the Greater Poland Voivodeship, and industrial processing and the production of textile products and electrical equipment are well developed as well. The total emissions of heavy metals to the atmosphere from the most harmful plants in the Greater Poland Voivodeship in 2015 were lower than in the Masovian Voivodeship: 0.006 t for Cd and 0.951 t for Pb [36]. The main center of the copper industry in Poland, i.e., the Legnica–Głogów Copper District (LGOM), is located in the vicinity of the Leszno hunting district. It is one of the largest centers of copper extraction in the world. The copper deposits located in this area are accompanied by silver as well as minerals of Pb, zinc, cobalt, molybdenum, nickel, selenium, rhenium, gold and platinum. There are also concentrates, sludge and dust containing compounds of Pb, Cd and other metals on the site of the Głogów and Legnica copper smelters [39].

The material for analysis was the breast muscles, leg muscles and liver of 28 wild mallard ducks, including 12 harvested in the Siedlce hunting district and 16 from the Leszno hunting district. The period during which the mallards were harvested was limited to the first two months of the hunting season for the species, i.e., from 15 August to 15 October, before the birds had begun to migrate. Detailed information on the methods of obtaining the mallard ducks can be found in the work by Bombik et al. [33]. The samples were frozen and stored at −20 °C.

The material for this study was obtained in accordance with the requirements of the National Ethics Committee for Animal Experiments of the European Union (authorization Nos. 37/2001 and 36/2011).

### 2.2. Laboratory Analysis

The concentrations of Pb and Cd were determined by inductively coupled plasma atomic emission spectroscopy (ICP OES) by using the PerkinElmer Optima 2000 DV spectrometer (USA) following the mineralization of the test material in the Anton Paar Microwave Extraction System (Austria). After thawing, the material was homogenized in an agate mortar. Weighted samples of about 1 g were transferred to quartz pressure vessels, to which 5.0 mL of 65% HNO_3_ (Suprapur^TM^, Merck, Rahway, NJ, USA) and 1 mL of 30% H_2_O_2_ (Suprapur^TM^, Merck) were added. The vessels were tightly sealed and placed in the mineralizer, which was equipped with a constant temperature- and pressure-control system. The cooled and degassed (CO_2_, NO_2_) digest was made up to 10 mL in class A volumetric flasks (BRAND). Radiation emissions were measured in the resulting solutions by selecting the longer axial (along the plasma) optical path for Pb and Cd. Calibration curves were prepared by using Merck ICP multielement standard solution XVI. The reagents used for mineralization were added to the standard solutions at the concentration remaining in the sample following mineralization in order to minimize potential interference during the introduction of the sample to the plasma and other physical interference. Analyses of each sample were performed in triplicate.

### 2.3. Statistics

Basic statistical measures were determined for each trait in each group, i.e., for the hunting districts and birds’ sex: the arithmetic mean ($\bar{x}$), range of variation (x_min_, x_max_), standard deviation (s) and coefficient of variation (V%).

In addition, a two-way nonorthogonal analysis of variance (Fisher–Snedecor F-test) with interaction was performed according to the following mathematical model:y_ijl_ = m + a_i_ + b_j_ + ab_ij_ + e_ijl_,
where

y_ijl_—the value of the trait for the ith hunting district (a = 2), jth sex (b = 2) and lth replicate (measurement).

m—the population mean.

a_i_, b_j_—the main effects of the factors, i.e., hunting district and sex.

ab_ij_—the effect of the interaction of hunting district and sex.

e_ijl_—the sampling error.

Significant effects were compared by using Tukey’s test at a significance level of 0.05.

Statistical analyses were performed by using Statistica 13.0 software.

## 3. Results

### 3.1. Content of Pb in Selected Tissues and Organs of Mallards (Anas platyrhynchos L.)

The results for the Pb content in the selected tissues and organs of the mallards (*Anas platyrhynchos* L.) are presented in Table 1 and Table 2.

The average accumulation of Pb in the breast muscle was higher, but not statistically significant, in males (0.0900 mg·kg^−1^ w.w.) than in females (0.0783 mg·kg^−1^ w.w.). Statistically significant differences in the average Pb concentration in this tissue were shown between the birds harvested in the Leszno hunting district (0.0986 mg·kg^−1^ w.w.) and individuals from the Siedlce hunting district (0.0649 mg·kg^−1^ w.w.). The coefficient of variation for the level of Pb in the breast muscle of the mallards from the two hunting districts was 19.71% for those from the Siedlce hunting district and 22.42% for those from the Leszno hunting district. In both hunting districts, the average level of Pb in the breast muscle was higher in the male mallards than in the females, but the differences were not statistically significant.

The average level of Pb in the leg muscles was higher in the male mallards (0.1341 mg·kg^−1^ w.w.) than in the females (0.0877 mg·kg^−1^ w.w.), although these were not statistically significant differences. The average Pb content in the leg muscles was higher in the mallards harvested in the Siedlce hunting district (0.1294 mg·kg^−1^ w.w.) than in the birds from the Leszno hunting district (0.0970 mg·kg^−1^ m.m.), although these were not statistically significant differences. In the Siedlce hunting district, a higher level of Pb was noted in the leg muscles of the males than in the females (Table 2), while the reverse relationship was noted in the Leszno hunting district. The differences were not statistically significant in either hunting district.

Permissible levels of Pb for other species were exceeded in the breast muscle of four mallards from the Leszno hunting district, while levels in the leg muscles were exceeded in seven samples (three in the Siedlce hunting district and four in the Leszno hunting district).

The average level of Pb in the liver of the mallards was higher in the males (0.1456 mg·kg^−1^ m.m.) than in the females (0.1219 mg·kg^−1^ w.w.), although the differences were not statistically significant. A significantly, more than two-fold higher average Pb content was noted in the liver of the mallards harvested in the Leszno hunting district (0.1758 mg·kg^−1^ w.w.) compared to the birds from the Siedlce hunting district (0.0778 mg·kg^−1^ w.w.). The coefficient of variation for the content of Pb in the liver of the birds from the hunting districts was 43.56% for those from the Siedlce hunting district and 56.55% for those from the Leszno hunting district. In the Siedlce hunting district, the average level of Pb in the liver was higher in the females than in the males. The reverse relationship was noted in the Leszno hunting district, but the differences were not statistically significant in either hunting district.

### 3.2. Content of Cd in Selected Tissues and Organs of Mallards (Anas platyrhynchos L.)

The results for the content of Cd in the selected tissues and organs of the mallards (*Anas platyrhynchos* L.) are presented in Table 3 and Table 4.

The average content of Cd in the breast muscle of the mallards was higher in the males (0.0179 mg·kg^−1^ w.w.) than in the females (0.0163 mg·kg^−1^ w.w.), but these differences were not statistically significant. The average level of this element in the breast muscle of the mallards was significantly higher in the ducks harvested in the Leszno hunting district (0.0189 mg·kg^−1^ w.w.) than in the birds from the Siedlce hunting district (0.0146 mg·kg^−1^ w.w.). In the birds harvested in the Leszno hunting district, the variation in the level of Cd in that tissue (21.28%) was twice as high as in the birds from the Siedlce hunting district (10.64%). In the Siedlce hunting district, the average level of Cd in the breast muscle of the female mallards was higher than in the males, while the reverse pattern was noted in the Leszno hunting district. These differences were not statistically significant in either hunting district.

The average Cd level in the leg muscles was higher in the female mallards (0.0174 mg·kg^−1^ w.w.) than in the males (0.0163 mg·kg^−1^ w.w.), although these differences were not statistically significant. The leg muscles of the mallards harvested in the Leszno hunting district had significantly higher average Cd content (0.0189 mg·kg^−1^ w.w.) than those of the birds from the Siedlce hunting district (0.0140 mg·kg^−1^ w.w.). The coefficient of variation for the content of the element in the leg muscles of the mallards from the hunting districts was 12.40% for those in the Siedlce hunting district and 23.29% for those in the Leszno hunting district. In both hunting districts, the average Cd concentration in the leg muscles was higher in the females than in the males, but these differences were not statistically significant.

The average level of Cd in the liver was higher in the males (0.0472 mg·kg^−1^ w.w.) than in the females (0.0357 mg·kg^−1^ w.w.), although these differences were not statistically significant. In the liver of the mallards harvested in the Siedlce hunting district, the average Cd concentration was significantly higher (0.0518 mg·kg^−1^ w.w.) than in the liver of the birds from the Leszno hunting district (0.0336 mg·kg^−1^ w.w.). The coefficient of variation for the content of Cd in the liver of the mallards was similar in both hunting districts: 45.34% in those from the Leszno hunting district and 47.52% in those from the Siedlce hunting district. In both the Siedlce and Leszno hunting district, the average Cd level was higher in the liver of the males than the females, although these differences were not statistically significant.

## 4. Discussion

In the present study, the average level of Pb in the breast muscle of mallard ducks in both analyzed hunting districts was lower than the levels reported by [40] and [6] for the same species in Warmia–Masuria. Szymczyk et al. [40] reported that the level of Pb in the breast muscle of mallards harvested in Warmia–Masuria ranged from 0.038 to 0.283 mg·kg^−1^ w.w. (on average 0.136 mg·kg^−1^ w.w.), while in birds harvested in Silesia, the range was 0.057 to 0.120 mg·kg^−1^ w.w. (on average 0.088 mg·kg^−1^ w.w.). According to the authors, the differences are explained by the significantly higher level of hunting exploitation in the former region. Hunting took place regularly in Warmia–Masuria, but only up to twice a year in Silesia. Zalewski et al. [6] reported Pb concentrations in the breast muscle of mallards ranging from 0.16 to 0.33 mg·kg^−1^ w.w. According to the authors, this concentration was influenced by the fact that the water body from which the birds were harvested was located near a sawmill, where there is generally more automobile traffic.

Sinkakarimi et al. [41] showed a higher average level of Pb in the breast muscle of mallards and common pochards wintering on the southeastern coast of the Caspian Sea compared to the levels obtained in the present study in the mallards from the two hunting districts. The average levels of Pb in the breast muscle of the mallards and common pochards reported by the authors were 0.83 mg/kg w.w. and 0.62 mg/kg w.w., respectively. Hutařová et al. [8], in a group of mallards kept in captivity without access to a water body in which they could swim, showed a lower concentration of Pb in the breast muscle than was shown in the mallards from the Leszno hunting district, whereas in a group of mallards harvested from the natural environment, the authors showed a higher level of Cd than in the breast muscle of the mallards from the Leszno hunting district. Hutařová et al. [8] showed a higher average Pb concentration in the breast muscle and liver of mallards (0.253 mg·kg^−1^ w.w. and 7.669 mg·kg^−1^ w.w., respectively) kept with access to a pond than in birds without access to a water body (0.077 mg·kg^−1^ w.w. and 0.287 mg·kg^−1^ w.w., respectively). According to the authors, the fact that the level of Pb in the tissues and organs of birds with access to a water body was several times as high may be explained by the intensive hunting exploitation of the study area and accumulation of Pb in the bottom layers of the water bodies where mallards forage. In this group of birds, Hutařová et al. [8] observed Pb concentrations exceeding the permissible levels in eight leg muscle samples and eight liver samples.

Plessl et al. [42] also showed a higher average level of this xenobiotic in the breast muscle of mallards harvested in eastern Austria compared to the present study. Their analysis of the levels of Pb in the tissues and organs of mallards revealed a higher average level in the liver (0.289 mg·kg^−1^ w.w.) than in the breast muscle (0.177 mg·kg^−1^ w.w.). However, the highest Pb concentration was detected in the breast muscle (7.89 mg·kg^−1^ w.w.). The high level of Pb in this tissue was due to contamination by Pb pellets during hunting.

Binkowski and Sawicka-Kapusta [43], in mallards and Eurasian coots harvested in the vicinity of Zator, and Dżugan et al. [44], in common pheasants shot near Rzeszów, showed lower levels of Pb in the breast muscle compared to the mallards obtained in the two hunting districts in the present study. Binkowski and Sawicka-Kapusta [43], in their analysis of Pb levels in the tissues and organs of mallards, noted higher levels in individuals obtained on fish ponds in the Zator area (40 km from Krakow) than in the vicinity of Milicz. The highest levels of Pb in the liver and breast muscle of mallards obtained near Zator were 31.72 mg·kg^−1^ d.w. and 18.96 mg·kg^−1^ d.w., respectively, while the corresponding levels in the vicinity of Milicz were 0.47 mg·kg^−1^ d.w. and 0.81 mg·kg^−1^ d.w., respectively. According to the authors, the high level of Pb in the tissues and organs of the mallards obtained from the fish ponds in Zator is due to intensive hunting exploitation in the area. Lower average concentrations of this xenobiotic in the liver in comparison to the present study were recorded by [40] in mallards from Warmia–Masuria; by [44] in game pheasants harvested in the vicinity of Rzeszów; and by [45] in farmed ostriches, turkeys and broiler chickens. Dżugan et al. [44], in the breast muscle and liver of game pheasants harvested in the vicinity of Rzeszów, showed average Pb levels of 0.007 mg·kg^−1^ d.w. and 0.059 mg·kg^−1^ d.w., respectively.

Ligocki and Dańczak [23] showed a higher average Pb level in the liver of male and female mallards compared to the levels obtained in the present study. A higher average concentration of this xenobiotic was also shown by [6] in the liver of mallards obtained in Warmia–Masuria, by [46] in individuals harvested in Kanibarazan in Iran, by [47] in birds from the southwest Atlantic coast in France, by [48] in birds from eastern Poland, by [42] in individuals harvested in eastern Austria and by [49] in birds from the vicinity of Lubartów in southeastern Poland. Alipour et al. [46] recorded an average Pb content in the liver of mallards amounting to 2.37 mg·kg^−1^ d.w., while [47], in the same species on the southwest Atlantic coast in France, detected an average level of 3.47 mg·kg^−1^ d.w. in the liver, which is indicative of the high level of Pb contamination of the Garonne and Dordogne estuaries. Sujak et al. [49] reported an average Pb concentration ranging from 0.52 to 0.88 mg·kg^−1^ d.w. in the liver of wild mallards harvested on fish ponds in Lubartów. Kalisińska et al. [15] noted a lower average level of this xenobiotic in the liver of juvenile and adult mallards harvested in the vicinity of Szczecin (0.228 mg·kg^−1^ w.w. and 0.229 mg·kg^−1^ w.w., respectively) compared to the levels obtained in the liver of the mallards from the Leszno hunting district.

The acceptable level of Pb was exceeded in the breast muscle of the mallards in nine samples and in the leg muscles in seven cases (35.71%). The level of Pb exceeding 0.1 mg/kg wet weight in these muscles may have been due to the small size of the mallard carcasses, and thus the short distance separating the site of the sampled tissue and the gunshot wounds. According to Zowczak et al. [50], when a Pb shot is used, Pb values in the muscles are highly varied and do not reflect the actual degree of contamination of the environment. On the other hand, it should be noted that the acceptable levels of Pb were exceeded mainly in the Leszno hunting district (100% in the breast muscle and 70% in the leg muscles). Given that emissions of this element to the atmosphere from the most harmful plants are lower in the Greater Poland Voivodeship than in the Masovian Voivodeship, this may indicate the impact of the use of phosphate fertilizers and fertilizer lime on the accumulation of Pb in the tissues and organs of mallards. Another potential source of contamination is the industrial plants in the Legnica–Głogów Copper District, exploiting deposits containing Pb minerals and located about 80 km away from the Leszno hunting district. In the present study, the average level of Pb in the liver of the mallards was higher than in the muscles, which is consistent with the previous findings of [8]. Kołacz et al. [51] showed that the copper industry does not affect the levels of lead and cadmium or the biochemical parameters in the blood of dairy cows. Kołacz et al. [52,53] found that with regard to Cu and Zn, there are currently no grounds to recognize the Żelazny Most landfill, located in the LGOM district, as a toxicological threat to the natural and agricultural environment. The results indicate that the livers of the mallards harvested in the Siedlce and Leszno hunting districts do not exceed the European limit for Pb levels in farm animals, amounting to 0.5 mg/kg. In contrast, ref. [48] recorded Pb levels exceeding acceptable limits in 8.6% of liver samples from this species. Guitart et al. [54] reported a similar level of Pb (in 9% of samples) in the liver of mallards harvested in northern Spain. Plessl et al. [42] showed that the acceptable level of this xenobiotic in the liver of mallards harvested in eastern Austria was exceeded in 3.9% of samples. Kim and Oh [10] reported Pb levels above 5 µg/g in the liver of 15% of mallards, 18.8% of Eurasian wigeons, 36.8% of spot-billed ducks and 50% of white-fronted geese.

Higher average levels of Cd in the breast muscle of mallards compared to the levels obtained in the birds from the two hunting districts in the present study were detected by Kalisińska et al. [15] in the breast muscle of adult mallards harvested in the Słońsk Nature Reserve, by [6] in mallards harvested in Warmia–Masuria and by [41] in mallards wintering on the southeastern coast of the Caspian Sea. Kalisińska et al. [15] recorded a higher average level of Cd in the liver of adult mallards (0.869 mg·kg^−1^ w.w.) than in juveniles (0.106 mg·kg^−1^ w.w.). The authors noted the same pattern in the liver of mallards harvested in the vicinity of Szczecin, but the Cd concentration in birds from this region was four times lower (0.282 mg·kg^−1^ w.w. in adult mallards and 0.028 mg·kg^−1^ w.w. in juveniles). According to the authors, the high level of this xenobiotic in the liver of the birds harvested in the Słońsk Nature Reserve was due to the regular deposition of pollutants in this area by the Oder River. Zalewski et al. [6], in their analysis of the concentrations of heavy metals in the breast muscle and liver of mallards, showed a lower average Cd content in the livers of juvenile birds (0.09 mg·kg^−1^ w.w.) than in older ones (0.16 mg·kg^−1^ w.w.). The authors draw attention to the fact that the highest Cd concentrations in both the breast muscle (0.27 mg·kg^−1^ w.w.) and liver (0.23 mg·kg^−1^ w.w.) were noted in birds harvested in the same areas. Sinkakarimi et al. [41] observed average concentrations of Cd in the breast muscle and liver of common pochards (0.59 mg·kg^−1^ w.w. and 1.63 mg·kg^−1^ w.w., respectively), which were more than twice as high as in mallards (0.21 mg·kg^−1^ w.w. and 0.69 mg·kg^−1^ w.w., respectively). According to the authors, these differences may be due to differences in how the birds forage, and thus in their food. Pochards feed on benthic organisms and therefore were at greater risk of exposure to Cd. Lower average concentrations of this xenobiotic in the breast muscle in comparison to the present study were reported by [42] in mallards harvested in eastern Austria and by [44] in game pheasants harvested in the vicinity of Rzeszów. Plessl et al. [42] showed a higher average level of Cd in the liver of mallards (0.228 mg·kg^−1^ w.w.) than in the breast muscle (0.002 mg·kg^−1^ w.w.). Dżugan et al. [44] also showed a 120-fold higher average level of Cd in the liver (0.721 mg·kg^−1^ m.m.) of game pheasants than in the breast muscle (0.006 mg·kg^−1^ w.w.). The liver and kidneys are commonly recognized as organs in which heavy metals, including Cd, are deposited [55].

Nikolić et al. [56] observed lower Cd concentrations in the leg muscles of mallards and game pheasants harvested in Serbia than were noted in the present study in the mallards harvested in both hunting districts. The authors tested Cd levels in the leg muscles and liver of game birds (common pheasant, mallard, Eurasian jay, grey partridge, Eurasian woodcock and common quail) harvested in Serbia in 2013–2016 and noted levels from <0.001 to 0.042 mg·kg^−1^ w.w. in the leg muscles and from 0.005 to 3.204 mg·kg^−1^ w.w. in the liver. They recorded the lowest average level of Cd in the leg muscles of grey partridge (0.004 mg·kg^−1^ w.w.) and the highest in Eurasian woodcock (0.042 mg·kg^−1^ w.w.). The lowest average Cd levels in the liver were noted in common quail (0.130 mg·kg^−1^ w.w.) and grey partridge (0.160 mg·kg^−1^ w.w.), and the highest levels were noted in Eurasian woodcock (1.247 mg·kg^−1^ w.w.). In mallards, the average Cd content in the breast muscle in that study was 0.005 mg·kg^−1^ m.m., and the level in the liver was 0.186 mg·kg^−1^ w.w.

Ligocki and Dańczak [23] observed a higher average Cd concentration in the liver of female long-tailed ducks than was shown in the present study in female mallards. The average concentration of Cd in the liver of female long-tailed ducks was twice as high (1.40 mg·kg^−1^ w.w.) as in the liver of female domestic ducks (0.74 mg·kg^−1^ w.w.). In the authors’ opinion, these differences may be more influenced by environmental factors than by the species of bird.

Felsmann et al. [22], in mallards harvested in the vicinity of Bydgoszcz and Żnin in 1995, found a lower average level of Cd in the liver than in the mallards harvested in the two hunting districts in the present study. This may be explained by the progressive anthropogenic pollution of the environment over the years and the resulting cumulative concentration of this xenobiotic in the liver of birds. Felsmann et al. [22] reported that the average Cd level in the liver of nesting mallards in 1993–1994 in the area adjacent to Bydgoszcz was 0.04 mg·kg^−1^ w.w., but in 1995 it decreased to 0.03 mg·kg^−1^ w.w., which may have been due to the reduction in production in nearby chemical plants and to shifting transit vehicular traffic from the city to the ring road. In birds nesting in the agricultural area in the vicinity of Żnin, the authors noted a doubling of the Cd level in the liver (from 0.03 mg·kg^−1^ w.w. to 0.06 mg·kg^−1^ w.w.). According to the authors, this was due to the fertilization of fields with phosphate fertilizer.

Szymczyk and Zalewski [40] showed a higher average level of Cd in the liver of mallards harvested in Warmia–Masuria than was shown in the birds harvested in the Leszno hunting district. Higher average Cd concentrations in the liver of mallards compared to the birds harvested in both hunting districts in the present study were reported by [6,46,48,49]. Dżugan et al. [44] also showed a higher average level of Cd in the liver of game pheasants harvested in the vicinity of Rzeszów. A lower average concentration of this xenobiotic in the liver of birds was reported by [42] in mallards harvested in eastern Austria. A lower average level of Cd in the liver of farmed poultry was reported by [45].

In the present study, the acceptable limits for Cd in farm animals (for muscle: 0.05 mg/kg w.w.; for liver: 0.5 mg/kg w.w.) were not exceeded in the muscles and liver of the mallards. Felsmann et al. [22] found that these limits were exceeded in the liver of 40% of the tested mallards nesting in the Żnin area. Binkowski et al. [24] observed excessive levels of this xenobiotic in the liver of 47% of mallards and Eurasian coots harvested in Zator. The higher Cd concentrations shown in the breast muscle and liver of mallards harvested on fish ponds in the vicinity of Zator (median: 0.9562 mg·kg^−1^ d.w. and 3.4486 mg·kg^−1^ d.w., respectively) may have been due to pollution of the area with household wastewater and the high levels of phosphorus fertilization of nearby crop fields. Plessl et al. [42] found that these limits were exceeded in six liver samples from mallards harvested in eastern Austria. Sujak et al. [49] reported that the acceptable Cd level was exceeded in the liver of mallards harvested on fish ponds in Lubartów in southeastern Poland in 4.3% of samples. The average Cd level in the liver of wild mallards ranged from 0.57 to 1.04 mg·kg^−1^ d.w.

Lucia et al. [47], in the liver of mallards harvested near the southwest Atlantic coast in France, showed a Cd level in the liver (14.79 mg·kg^−1^ d.w.) that was 14 times that reported by Alipour et al. [46] in this species (1.05 mg·kg^−1^ d.w.), which may have been due to that fact that the location where the mallards lived and were harvested was severely contaminated with this xenobiotic.

In the present study, among all the tested tissues and organs from the mallards, the highest average Pb concentration was noted in the liver, and it was significantly higher in the birds harvested in the Leszno hunting district compared to the Siedlce hunting district. This high average level of Pb in the liver of birds from the Leszno hunting district may have been due to the intensive hunting exploitation of the region, the deposition of Pb together with phosphorus fertilizer and fertilizer lime and point emissions of particulate pollution from the extraction of deposits containing Pb minerals in the Legnica–Głogów Copper District. Acceptable levels of Pb for other species were exceeded in the mallards in four breast muscle samples from the Leszno hunting district and in seven leg muscle samples (three in the Siedlce hunting district and four in the Leszno hunting district). The results may reflect the level of this xenobiotic in the environment and may also have been due to the contamination of the tissues with a Pb shot, owing to the small size of the mallard carcasses.

The highest average Cd concentration was recorded in the breast and leg muscles of the mallards harvested in the Leszno hunting district and in the liver of the birds from the Leszno hunting district. The concentrations of Cd in the tissues and organs of the mallards harvested in the Siedlce or the Leszno hunting district did not exceed acceptable limits for the muscles and liver of slaughtered animals.

This study showed no significant effect of the sex of the wild mallards on the content of lead and cadmium in their tissues and organs.

## 5. Conclusions

This study showed that the mallard has a measurable response to environmental pollution with Pb and Cd, and therefore it is a species that could to some extent be used as a bioindicator of the level of contamination of the environment with these xenobiotics. The ban on the use of Pb pellets as ammunition in Poland may largely limit the degree of Pb contamination of the tissues and organs of mallards. Further research will be needed to determine how the elimination of this environmental contaminant will affect the lead content of waterfowl tissues and organs.

## Figures and Tables

**Table 1 animals-13-03327-t001:** Average Pb content (mg·kg^−1^ w.w.) in selected tissues and organs of mallards (*Anas platyrhynchos* L.) by sex and hunting district.

Tissues and Organs	Basic Statistics	Sex		LSD_0.05_	Hunting District		LSD_0.05_
		Female (*n* = 14)	Male (*n* = 14)		Siedlce (*n* = 12)	Leszno (*n* = 16)	
Breast muscles mg·kg^−1^ w.w.	$\bar{x}$	0.0783 a	0.0900 a	n. s.	0.0649 a	0.0986 b	0.0149
	min.	0.0459	0.0506		0.0459	0.0563	
	max.	0.1082	0.1382		0.0980	0.1382	
	s	0.0195	0.0284		0.0128	0.0221	
	V%	24.87	31.61		19.71	22.42	
Leg muscles mg·kg^−1^ w.w.	$\bar{x}$	0.0877 a	0.1341 a	n. s.	0.1294 a	0.0970 a	n. s.
	min.	0.0412	0.0530		0.0412	0.0623	
	max.	0.1801	0.7683		0.7683	0.1801	
	s	0.0352	0.1770		0.1941	0.0272	
	V%	40.11	132.03		149.95	28.05	
Liver mg·kg^−1^ w.w.	$\bar{x}$	0.1219 a	0.1456 a	n. s.	0.0778 a	0.1758 b	0.0640
	min.	0.0527	0.0559		0.0527	0.0858	
	max.	0.3452	0.4401		0.1806	0.4401	
	s	0.0713	0.1078		0.0339	0.0994	
	V%	58.50	74.00		43.56	56.55	

Legend: $\bar{x}$—arithmetic mean; min., max.—extreme values; s—standard deviation; V%—coefficient of variation; LSD_0.05_—least significant difference (*p* ≤ 0.05); a, b—means marked with different letters for sex and hunting district are significantly different (*p* ≤ 0.05). n. s.—no significance.

**Table 2 animals-13-03327-t002:** Average Pb content (mg·kg^−1^ w.w.) in selected tissues and organs of mallards (*Anas platyrhynchos* L.) for the sexes in the hunting districts.

Tissues and Organs	Basic Statistics	Hunting District				LSD_0.05_
		Siedlce		Leszno		
		Female (*n* = 6)	Male (*n* = 6)	Female (*n* = 8)	Male (*n* = 8)	
Breast muscles mg·kg^−1^ w.w.	$\bar{x}$	0.0623 a	0.0674 a	0.0903 a	0.1069 a	n. s.
	min.	0.0459	0.0506	0.0563	0.0740	
	max.	0.0722	0.0980	0.1082	0.1382	
	s	0.0094	0.0150	0.0162	0.0240	
	V%	15.00	22.32	17.92	22.50	
Leg muscles mg·kg^−1^ w.w.	$\bar{x}$	0.0697 a	0.1891 a	0.1012 a	0.0928 a	n. s.
	min.	0.0412	0.0530	0.0623	0.0628	
	max.	0.1113	0.7683	0.1801	0.1164	
	s	0.0270	0.2597	0.0346	0.0158	
	V%	38.75	137.36	34.14	17.08	
Liver mg·kg^−1^ w.w.	$\bar{x}$	0.0907 a	0.0648 a	0.1453 a	0.2063 a	n. s.
	min.	0.0527	0.0559	0.0856	0.0905	
	max.	0.1806	0.0797	0.3452	0.4401	
	s	0.0436	0.0077	0.0787	0.1082	
	V%	48.06	11.87	54.19	52.45	

Legend: $\bar{x}$—arithmetic mean; min., max.—extreme values; s—standard deviation; V%—coefficient of variation; LSD_0.05_—least significant difference (*p* ≤ 0.05); a—means marked with different letters for sex and hunting district are significantly different (*p* ≤ 0.05). n. s.—no significance.

**Table 3 animals-13-03327-t003:** Average Cd content (mg·kg^−1^ w.w.) in selected tissues and organs of mallards (*Anas platyrhynchos* L.) by sex and hunting district.

Tissues and Organs	Basic Statistics	Sex		LSD_0.05_	Hunting District		LSD_0.05_
		Female (*n* = 14)	Male (*n* = 14)		Siedlce (*n* = 12)	Leszno (*n* = 16)	
Breast muscles mg·kg^−1^ w.w.	$\bar{x}$	0.0163 a	0.0179 a	n. s.	0.0146 a	0.0189 b	0.0020
	min.	0.0132	0.0116		0.0116	0.0132	
	max.	0.0200	0.0291		0.0179	0.0291	
	s	0.0022	0.0049		0.0016	0.0040	
	V%	13.62	27.21		10.64	21.28	
Leg muscles mg·kg^−1^ w.w.	$\bar{x}$	0.0174 a	0.0163 a	n. s.	0.0140 a	0.0189 b	0.0030
	min.	0.0121	0.0108		0.0108	0.0121	
	max.	0.0324	0.0224		0.0185	0.0324	
	s	0.0052	0.0030		0.0017	0.0044	
	V%	29.90	18.36		12.40	23.29	
Liver mg·kg^−1^ w.w.	$\bar{x}$	0.0357 a	0.0472 a	n. s.	0.0518 b	0.0336 a	0.0160
	min.	0.0165	0.0243		0.0165	0.0201	
	max.	0.0953	0.0862		0.0953	0.0862	
	s	0.0204	0.0216		0.0246	0.0152	
	V%	57.12	45.84		47.52	45.34	

Legend: $\bar{x}$—arithmetic mean; min., max.—extreme values; s—standard deviation; V%—coefficient of variation; LSD_0.05_—least significant difference (*p* ≤ 0.05); a, b—means marked with different letters for sex and hunting district are significantly different (*p* ≤ 0.05). n. s.—no significance.

**Table 4 animals-13-03327-t004:** Average Cd content (mg·kg^−1^ w.w.) in selected tissues and organs of mallards (*Anas platyrhynchos* L.) for the sexes in the hunting districts.

Tissues and Organs	Basic Statistics	Hunting District				LSD_0.05_
		Siedlce		Leszno		
		Female (*n* = 6)	Male (*n* = 6)	Female (*n* = 8)	Male (*n* = 8)	
Breast muscles mg·kg^−1^ w.w.	$\bar{x}$	0.0150 a	0.0142 a	0.0172 a	0.0206 a	n. s.
	min.	0.0135	0.0116	0.0132	0.0145	
	max.	0.0179	0.0162	0.0200	0.0291	
	s	0.0015	0.0015	0.0022	0.0047	
	V%	9.76	10.73	12.89	22.57	
Leg muscles mg·kg^−1^ w.w.	$\bar{x}$	0.0144 a	0.0137 a	0.0197 a	0.0182 a	n. s.
	min.	0.0123	0.0108	0.0121	0.0152	
	max.	0.0185	0.0154	0.0324	0.2243	
	s	0.0020	0.0014	0.0057	0.0023	
	V%	13.70	10.22	28.89	12.87	
Liver mg·kg^−1^ w.w.	$\bar{x}$	0.0447 a	0.0590 a	0.0290 a	0.0383 a	n. s.
	min.	0.0165	0.0243	0.0201	0.0247	
	max.	0.0953	0.0849	0.0443	0.0862	
	s	0.0275	0.0188	0.0073	0.0192	
	V%	61.62	31.84	25.24	50.13	

Legend: $\bar{x}$—arithmetic mean; min., max.—extreme values; s—standard deviation; V%—coefficient of variation; LSD_0.05_—least significant difference (*p* ≤ 0.05); a—means marked with different letters for sex and hunting district are significantly different (*p* ≤ 0.05). n. s.—no significance.

## Data Availability

No new data were created or analyzed in this study. Data sharing is not applicable to this article.

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
