# Peer review of "Content of Lead and Cadmium in the Tissues and Organs of the Wild Mallard Duck (Anas platyrhynchos L.) Depending on the Region of Poland Where It Is Harvested"

_animals, 2023, doi:10.3390/ani13213327_

Round 1

Reviewer 1 Report

In this paper, Bombik et al analyze Cd and Pb content in tissues and organs of the wild mallard duck in order to assess the degree of environmental pollution in some regions of Poland.

The study is well structured, however, in order to be improved the authors should make the following changes:

1) In the text the words "Cadmium" and "lead" are used alternating "Cd" and "Pb". The authors should choose a single nomenclature and use it throughout the text.

2) The statistical analysis were well conducted and accurate. The authors should include summary graphs to better understand the differences in contaminant content between animal groups

The quality of English are appropriate.

Author Response

Thank you very much for your review. We provide: answers:

  1. The words “cadmium” and “lead” in the text have been changed to “Cd” and “Pb”.
  2. The data presented in the tables are complete and sufficient. There is no need to present the data in graphs, because the data concern main effects (means, extreme values, coefficients of variation, and LSD)

Reviewer 2 Report

The manuscript "Content of lead and cadmium in the tissues and organs of the wild mallard duck (Anas platyrhynchos L.) depending on the region of Poland where it is harvested" describes scientific research conducted on the analysis of the content of selected heavy metals in the tissues of wild mallard ducks. Tissues of animals included in the research were obtained in two locations in Poland, and the collection of research material was random - the analyzes were performed on birds obtained during routine hunting procedures.

Present topic is particularly important in the assessment of environmental contamination and the evaluation of the content and bioaccumulation of various trace elements in wild animals.

The literature review included in the introduction to the article was prepared appropriately. The Authors made a detailed analysis of literature data, including numerous national and international information. Taking into account the largest European post-flotation waste reservoir, which the authors also refer in the text (Legnica-GÅ‚ogów Copper District), it would be worth to include in the manuscript also the references to heavy metal researches performed earlier in this area.
I strongly recommend:

1) https://www.webofscience.com/wos/woscc/full-record/WOS:000395368600007

(http://www.medycynawet.edu.pl/index.php/archives/396/5652-summary-med-weter-73-3-171-175-2017);

2) https://www.webofscience.com/wos/woscc/full-record/WOS:000985714100001

(https://jsite.uwm.edu.pl/articles/view/1976/)

3) https://www.webofscience.com/wos/woscc/full-record/WOS:000343689800041

The methodological part was described in a logical and fully understandable way. Due to the bioindication argument rightly pointed by the Authors, apart from the described period material collection (i.e. day and month), it is also worth to supplement the year of analyzes carried out. This information will be valuable when comparing data by other researchers who may have data from other calendar years.

The statistical tests were selected appropriately, which allowed for precise presentation of the results, as well as comparison of the data obtained in two different locations.

The extensive discussion and its insight into the conducted research are a valuable culmination of the manuscript. Noteworthy is the pursuit of an objective assessment of own research, which clearly indicates the high level of the authors and the analyzes performed.

Without a doubt I believe that this manuscript is a valuable scientific work which fully deserves for publication in "Animals", what I strongly recommend.

Author Response

Thank you very much for your review. We provide: answers:

References to previous research on heavy metals in the LGOM district have been added to the manuscript.

The years of the study have been given.

Reviewer 3 Report

Summary

The aim of this study was to determine the effect of hunting district and sex on the content of lead and cadmium in the braest and leg muscle and liver of the wild mallard duck (Anas platyrhynchos L.). The research results supplement the knowledge about the concentration of lead and cadmium in the muscle tissue and liver of wild game ducks of the species Anas platyrhynchos L., which contributes to the current state of knowledge.

General concept comments.

The Introduction chapter provides an overview of the world's knowledge on this subject. In my opinion, this chapter should be supplemented with information on the species of wild ducks classified as game birds in Poland, the size of the population of wild ducks in Poland and the number of shootings (see Statistical Yearbook of Hunting, etc.), Is there any farm breeding of wild ducks in Poland? Advantages and risks of eating wild duck meat. The material used in the research is sufficiently numerous, and the research methods used are correct. The discussion is exhaustive, although an explanation of the basis for the differences between lead and cadmium content between males and females is required. Summary of the results are correct. The abstract chapter requires supplementation and correction in several places.

Specific comments

L 21 The higher….

  L22 "...in the Leszno hunting district than Siedlce hunting district." Instead of current form

L25 No description of significant differences in lead concentration in breast muscles and liver depending on the hunting district.

L58 Waterbird or Waterfowl?

L141 In what part of Poland is the Leszno hunting district?

L145 In what part of Poland is the Siedlce hunting district?

L256 Geater Poland "Voivodeship"?

L180 Bombik et al. [32] - instead of current form

In Tables 1-4 "Breast muscles..." not bold

L253 22.6-fold higher?

L264 Table 3 from the next page

L396 “Sinkarimi et al. [15]” - instead of current form

L478 Alipour et al. [45] - instead of current form

Compliance preparing an article according to editorial requirements

The references chapter change are nedded: volume in italic, year in bold, for page ranges use long (-) from the symbol function, instead of short (-) from the keyboard

Author Response

Thank you very much for your review. We provide: answers:

Information on species of wild ducks included among game birds in Poland and the number of wild duck shootings in Poland has been added to the Introduction. Information on farm breeding of wild ducks has been included as well. The advantages and risks of eating wild duck meat are discussed.

L 21, L22, L25, L58, L141, L145, L256, and L180 have been corrected.

In Tables 1-4 "Breast muscles..." is unbolded in accordance with the requirements of the journal.

L253 corrected

L264 corrected

L396 corrected

L478 corrected